# Preclinical Models to Study the Molecular Pathophysiology of Meniere’s Disease: A Pathway to Gene Therapy

**DOI:** 10.3390/jcm14051427

**Published:** 2025-02-20

**Authors:** Prathamesh T. Nadar-Ponniah, Jose A. Lopez-Escamez

**Affiliations:** 1Meniere Disease Neuroscience Research Program, Faculty of Medicine & Health, School of Medical Sciences, The Kolling Institute, University of Sydney, Sydney, NSW 2065, Australia; 2Otology & Neurotology Group CTS495, Division of Otolaryngology, Department of Surgery, Instituto de Investigación Biosanitaria, ibs.GRANADA, Universidad de Granada, 18071 Granada, Spain; 3Sensorineural Pathology Programme, Centro de Investigación Biomédica en Red en Enfermedades Raras, CIBERER, 28029 Madrid, Spain

**Keywords:** hearing loss, Meniere’s disease, deafness, vertigo, zebrafish, Drosophila, genetics

## Abstract

**Background:** Meniere’s disease (MD) is a set of rare disorders that affects >4 million people worldwide. Individuals with MD suffer from episodes of vertigo associated with fluctuating sensorineural hearing loss and tinnitus. Hearing loss can involve one or both ears. Over 10% of the reported cases are observed in families, suggesting its significant genetic contribution. The condition is polygenic with >20 genes, and several patterns of inheritance have been reported, including autosomal dominant, autosomal recessive, and digenic inheritance across multiple MD families. Preclinical research using animal models has been an indispensable tool for studying the neurophysiology of the auditory and vestibular systems and to get a better understanding of the functional role of genes that are involved in the hearing and vestibular dysfunction. While mouse models are the most used preclinical model, this review analyzes alternative animal and non-animal models that can be used to study MD genes. **Methods:** A literature search of the 21 genes reported for familial MD and the preclinical models used to investigate their functional role was performed. **Results:** Comparing the homology of proteins encoded by these genes to other model organisms revealed Drosophila and zebrafish as cost-effective models to screen multiple genes and study the pathophysiology of MD. **Conclusions:** Murine models are preferred for a quantitative neurophysiological assessment of hearing and vestibular functions to develop drug or gene therapy.

## 1. Introduction

Meniere’s disease (MD) is a debilitating inner ear disorder affecting around 20,000 Australians and over 4 million people worldwide. The disease is characterized by fluctuating sensorineural hearing loss (SNHL) associated with tinnitus, aural fullness (feeling pressure in the ear), and recurrent vertigo lasting for hours. Most of the individuals diagnosed with MD in due course progress to moderate to severe SNHL in the affected ear, with some getting persistent and disabling tinnitus. Those involving both ears (bilateral MD), which accounts for up to 40% of cases, also suffer from chronic imbalance that has a great impact on personal mental health and daily activities [1,2].

The condition is characterized by an accumulation of endolymph, termed endolymphatic hydrops (EH), in the cochlear duct (cochlea and vestibular end organs), and it is associated with immune-related disorders, allergy, and autoinflammation [3,4,5,6].

Due to its heterogenous nature, the exact cause of MD is still unknown, but multiple factors are known to contribute to the disease. For instance, environmental factors, such as diet or allergies, could contribute to the disease, but genetic factors and the immune response have a pivotal role in the MD pathophysiology [7]. While most of the cases are sporadic, there are numerous cases where family members show similar symptoms. MD exhibits a significant familial aggregation, with a sibling recurrence risk ratio ranging from 16 to 48, suggesting a strong genetic component [8]. These cases account for around 10% of the patients and are defined as familial Meniere‘s disease (FMD).

The first case of FMD was reported by Dr. Madeleine Ray Brown in 1941, observed within two families. The first family was a French-Canadian consanguineous family where two sisters and their brother suffered from SNHL and paroxysmal vertigo. The age of onset was between 32 and 46, with all members recorded to have had tinnitus followed by vertigo attacks [9]. The second family comprised two identical twins exhibiting audio-vestibular symptoms, with the first one showing non-progressive hearing loss, while the second had paroxysmal attacks of vertigo and a sudden increase in deafness and tinnitus from the age of 31. The early onset and association of audiological symptoms with the episodes of vertigo are essential features in FMD. In the subsequent years, other researchers have reported familial cases of MD, such as Bernstein [10], Morrison [11], Lee [12], and Requena [8]. It is interesting to note that most reported cases are sporadic, where relatives of MD patients show clinical differences in the age of onset, hearing profile, or vestibular involvement, leading to missing diagnoses or partial syndromes [13]. However, FMD accounts for 5–20% of reported cases in populations of European descent [14].

Preclinical models have been indispensable to study the pathophysiology of neurological diseases. To understand the complex pathophysiology of MD and the underlying molecular mechanism of the disease, many preclinical models have been used [15]. While many experimental animal models have been previously used to study MD, such as infusion of systemic vasopressin, or endolymphatic sac ablation [16,17,18], these models do not recapitulate the molecular biology underlying the origin or progression of the disease; in fact, they are models of EH and represent the end stage of the damage in the cochlear duct associated with the condition, as well as permanent hearing impairment [19]. Furthermore, EH is associated with MD and other inner ear disorders [20]. Research has found an association between hydrops and sensorineural hearing loss in humans, and rodent models are needed if the main goal is to assess EH. To overcome this shortfall, it would be ideal to generate preclinical models with identical mutations found in patients with MD to mimic the phenotype of the disease. This approach will allow us to establish a causal model triggered by specific genetic mutations associated with fragile proteins (i.e., Otogelin) to get a better understanding of the molecular biology of MD in the early stages.

## 2. Non-Animal Preclinical Models to Study MD

The potential application of non-animal models to investigate the molecular mechanisms of SNHL and MD has been recently reviewed by Lamolda et al. [21], but it is not the main goal of this perspective. Gene expression studies in human inner ear organoids are an excellent resource to investigate the expression pattern of existing or novel candidate genes for MD. Human inner ear organoids scRNAseq, snRNAseq, and single-cell mouse RNAseq datasets are accessible from gEAR (http://umgear.org acessed on 3 December 2024) at different ages (embryonic (E) 16, postnatal (P) 1, P7, P15, and P30). This information is critical to understand the cell types involved and the functional role of the candidate genes during otic capsule development. Figure 1 shows the gene expression profile in human inner ear organoids of the three most common genes associated with familial MD (*OTOG*, *MYO7A*, and *TECTA*) [22]. Otogelin, encoded by the *OTOG* gene, is present in the tectorial and otolithic membranes, contributing to the attachment of outer hair cells, stereocilia, to the acellular structures overlying the sensory epithelium [23,24]. Additionally, they are also expressed in the vestibular epithelial cells. Myosin VIIA protein encoded by the gene *MYO7A* is predominantly expressed in hair cells and vestibular epithelial cells. This expression suggests how important these proteins are for proper hearing. Alpha-tectorin is encoded by the gene *TECTA*. Similar to *MYO7A* and *OTOG*, it is also expressed in the cochlear duct floor cells and transitional or endolymphatic cells, and there is also moderate expression in the hair cells.

For decades, murine models have been the preferred model to study auditory neuroscience. In this review, we focus on animal and non-animal models that could be used to study auditory and vestibular systems, with a focus on FMD. Different gene mutations and types of inheritance have emerged in the last 10 years, including autosomal dominant, autosomal recessive, and digenic familial MD [1]. We will focus on these reported genes, the cell types involved in the cochlea and vestibular organs, and the most suitable animal model to study the biology of these genes in MD disease.

## 3. Animal Preclinical Models to Study MD

### 3.1. Drosophila as a Model to Study Genetics of Hearing Loss

While it sounds like a farfetched idea to use *Drosophila (Drosophila melanogaster)* as a model organism to study hereditary hearing loss in humans, intriguingly, the fruit fly shares analogous accessory structures to capture and transduce sound signals, similar to vertebrates, and they use analogous systems for mechanotransduction. In mammals, mechanosensitive cells (hair cells) found in the organ of Corti (hearing organ) of the inner ear serve as mechanoreceptors. They respond to sound waves and release neurotransmitters, leading to activation of neurons in the auditory pathway. Similarly, the hearing organ in fruit fly is the Johnston’s organ (as illustrated in Figure 2), which also contains several hundred mechanosensitive units, called the scolopidia, and is found in the second segment of the antenna [25]. Sound stimuli, such as courtship song, lead to bulk air displacements, causing rotation of distal antennal segments, thereby stretching the scolopidia [26]. The planar cell polarity (PCP) pathway, which dictates uniform arrangement of stereocilia on the apical surfaces of the sensory hair cells, is highly conserved between mammals and *Drosophila* [27]. There are substantial bodies of research that illustrate that the mammalian auditory system and Johnston’s organ are functionally similar [28,29,30,31]. Even though *Drosophila* and the vertebrates (mammals) are separated by several million years during evolution, the hearing organs are specified by a similar set of transcription factors, and they show numerous similarities with respect to proteins and accessory cells, which allow receptor cells to be mechanically gated [28]. For instance, Atoh1, which in vertebrates is important for differentiation and survival of the sensory hair cells, has a *Drosophila* orthologue (ato), which is required for the formation of mechanoreceptors and photoreceptors [32,33]. Furthermore, loss of ato/Atoh1 leads to loss of chordotonal organs, which includes Johnston’s organ in *Drosophila* [32], or loss of hair cell differentiation and survival in the mouse organ of Corti [34]. There are additionally mechanisms and gene expression patterns that both *Drosophila* and mice share, making it an interesting candidate to study genetics of hearing loss.

*Drosophila* as a Meniere’s disease model
**Strengths**

**Limitations**

Fast generation of knockout/knock-in or overexpression of the gene of interest [36].

While identifying a mammalian homolog in *Drosophila* seems simple, not all genes have the same function in the sensorial organs across different species [28].

Short life span and armory of genetic tools that can be used to manipulate gene expression [28].

Overexpression of a gene of interest can lead to artifacts [28].

Several hearing loss genes (i.e., Myosin VIIA) are highly conserved in *Drosophila* [29,37].

The functional role of the mutation or overexpression of a deafness gene in *Drosophila* may lead to a different consequence, and extrapolating its role in pathogenicity in humans is difficult [28].

Several genes responsible for genetic diseases (Usher syndrome) leading to hearing loss are highly conserved in *Drosophila* [28].

Limited frequency range: the mammalian hearing organ responds to a wide range of sound frequencies; however, *drosophila* only responds to a narrow frequency range of beating male wings [28].

Moreover, noise-induced hearing loss can be studied in *Drosophila* [38].


No need for animal ethical approval.



### 3.2. Zebrafish as a Model to Study Genetics of Hearing Loss

In the late 1970s, when an alternate model organism to study vertebrate gene function and development was being actively pursued, George Streisinger and his colleagues established the zebrafish, *Danio rerio*, model [39,40]. The zebrafish hearing organ comprises two hair cell sensory systems—the inner ear and lateral line (as illustrated in Figure 3).

The zebrafish has several resemblances to the mammalian hearing organ, illustrating its ideal candidature as a model organism to study the auditory system. For instance, the inner ear of zebrafish, similar to the mammalian inner ear, contains three semicircular canals and associated cristae. Furthermore, the two maculae in the zebrafish are also required for hearing and balance. The macula contains calcium carbonate crystals, collectively called otoliths [41,42]. In both larval as well as adult zebrafish, the utricular macula plays a crucial role in balance and detecting gravity [43]. Moreover, at the cellular and molecular levels, mammalian and zebrafish hair cells (HCs) are strikingly similar, with several genes expressed in HCs that are required for hearing and balance in both zebrafish and mammals (human and mouse) [44,45,46]. Both mammalian and zebrafish hearing organs are capable of detecting sound signals at low frequency. Physiological properties of zebrafish HCs using a patch clamp revealed, to a certain extent, to be like immature mammalian vestibular and auditory systems [47]. Interestingly, zebrafish are more sensitive to sound at lower frequency, making them a potential candidate to study genes associated with low-frequency hearing loss [48]. Despite of all these similarities, there is one major difference in the hearing organs between zebrafish and mammals, which is the saccule. While in zebrafish it is used to detect sound signals, in mammals this is performed by a specialized auditory organ, the organ of Corti, located in the anterior labyrinth [49]. Overall, forward genetic screens in zebrafish have provided and continue to provide a wealth of knowledge in the field of auditory science.

**Figure 3 jcm-14-01427-f003:**
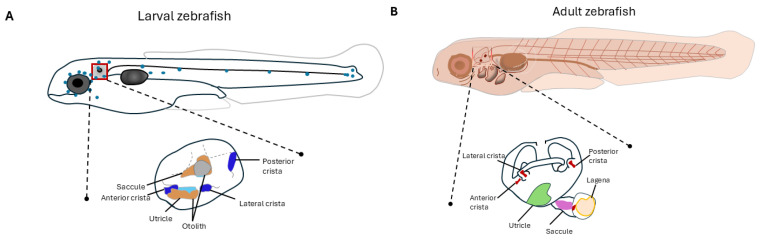
Hearing and balance organs of the zebrafish (**A**). Illustration of the lateral line system of larval zebrafish (5 dpf). Individual neuromasts (blue circles) are located on the body of the fish. The saccule, in soft brown, used to detect sound in larval zebrafish. The cristae are shaded in blue and macular sensory organs are colored brown and grey. (**B**) Illustrations of the adult zebrafish. Shown in soft brown color is the lagena (used to detect sound signal). The green and purple are macula otolith organs. The semicircular canal crista ampullaris are shaded in red (*Danio rerio*; Figure adapted from [50]).

Zebrafish as a Meniere’s disease model
**Strength**

**Limitations**

Highly accessible for in vivo study. During their development (embryonic and larval stages), zebrafish are transparent, enabling researchers to perform experimentation and observation in vivo [51]. This is a stark contrast to mammals, where the sensory epithelia are encapsulated in the temporal bone of the skull, making difficult the access for in vivo study.

Choice of appropriate transgenic lines is important. Some studies [52] have shown that some lines (Brn3c, Bcl2, and ET4) are less sensitive to auditory stimuli, as compared to others (Myo6).

They are beneficial for developmental studies due to the fast turn out time. It takes 5 days post-fertilization for zebrafish to develop hair cell sensory systems, making this an ideal study development pathway of genes in a short span of time [53].

Given that the fish are responding to the “sound” that is being transmitted in the water, which is actually particle motion, it is hard to know if the fish are responding to the sound from the microphone, sound pressure, or uncalibrated particle motion due to the configuration of the fish tank [54].

Zebrafish transgenic lines are easy to manipulate and can be augmented to suit experimental requirements. The zebrafish transgenic lines can be augmented to express florescent protein that helps in the visualization and study of hair cell sensory systems in a more sophisticated way [52].


Zebrafish forward genetic screens can be used to identify genes essential for hearing and balance [53].


Zebrafish are used for drug screening for otoprotective compounds and, for this reason, this could be an excellent MD model to test novel drugs [55].



## 4. Genes Linked to Autosomal Dominant Familial MD

Ten genes have been reported in several families with dominant inheritance: *FAM136A*, *DTNA*, *PRKCB*, *COCH*, *DPT*, *SEMA3D*, *TECTA*, *GUSB*, *SLC6A7,* and *GJD3*.
***FAM136A***


The gene *FAM136A* (OMIM: 616275) encodes the mitochondrial protein (family with sequence similarity 136 member A; Q96C01). It is expressed in the neurosensorial epithelium of the *crista ampullaris* and human lymphoblast cells. The protein is highly conserved across species and may be essential for oxidative phosphorylation. Using exome sequencing, it was found that three women from a Spanish family with the MD phenotype had a novel heterozygous variant in the *FAM136A* gene [56]. The frameshift mutation led to a stop codon (GRCh38 chr2: 70300842G > A; NM_032822.3), with a reduction in several transcripts by nonsense-mediated decay.

When the homology of the FAM136A protein sequence was compared with other animal models (Figure 4), the analysis revealed that after the mouse model, the most suitable model would be the guinea pig. Surprisingly, the zebrafish ortholog also shared homology of 66.9% with the human *FAM136A*. This suggests that the zebrafish model would be a viable alternate option to study the biology of the FAM136A.
***DTNA***

The human protein α-dystrobrevin (Q9Y4J8), encoded by the gene *DTNA* (OMIM: 601239), belongs to the dystrobrevin family. It is a structural component of the dystrophin–glycoprotein complex (DGC). It remains unclear what its role in inner ear function is but, due to its expression in the vestibular sensory epithelia during early development in mice [57], it has been suggested to have a pivotal role in maturation of the vestibular system. Furthermore, *DTNA* is expressed in several cell types in the organ of Corti (Deiter cells, outer hair cells, and pillar cells) and stria vascularis (marginal and intermediate cells) [58].

Sequencing analysis of three patients with MD who had a stop codon mutation in *FAM136A* showed that they also had a missense mutation for *DTNA* in chr18: 32462094G > T [56]. Comparing α-dystrobrevin homology of DTNA among different animal models, the guinea pig showed the highest similarity (96.7%), followed by the chinchilla (96.5%).

Of note, Requena et al. (2023) also proposed Drosophila as a model to study MD [59]. Research found that the Drosophila homologues of the DTNA, dystrobrevin (Dyb), and dystrophin (Dys) are expressed in the Johnston’s organ (JO). Although Dyb protein homology is only 47.7%, the authors showed that dystrobrevin is required in hearing and proprioception in Drosophila [59]. The study showed that the mutant fly had not only defective proprioceptive ability, but also an auditory homeostasis deficit, and exhibited hearing loss. These results support the use of Drosophila as an attractive model to study hearing and vestibular systems for mutations in *DTNA*. However, the zebrafish orthologue of *DTNA*, Dyb, shares a protein homology of 66.4% (Figure 4). Hence, the zebrafish model would be the preferred model to study the biology of MD for *DTNA.*
***PRKCB***

The protein kinase C beta type is an ATP-binding protein (P05771) encoded by the *PRKCB* gene (OMIM 176970) with varied cellular function, which has been implicated in MD. Two Spanish individuals affected with MD exhibited a novel missense variant in *PRKCB*. Using whole-exome sequencing and Sanger sequencing, it was revealed that the missense variant (chr16: 23999898G > T) in the *PRKCB* gene affects the protein sequence, and possibly its function. Patients with this mutation had low-frequency SNHL. The protein is expressed in a tonotopic gradient in the non-sensory cells (tectal cells and inner border cells) of the rat organ of Corti [60]. Comparing the protein sequence of *PRKCB* with other model organisms (Figure 4), chinchilla would be a preferred model due to its higher similarity to the human *PRKCB*. However, it is interesting to note that its zebrafish homologue also showed significant similarity (87%), suggesting it as an alternative model to study the *PRKCB* mutation in MD.
***COCH***

The gene *COCH* (OMIM 603196) encodes for the structural protein Cochlin (O43405), which is differently expressed in the inner ear. It is predominantly expressed in the lateral wall, outer sulcus cells, spiral ligament, and spiral limbus of the cochlea. Being a structural protein, it is essential for the maintenance of structural integrity and architecture of the lateral wall of the cochlea. In the vestibular system, it is the second most frequent protein to be expressed [61].

Whole-exome sequencing (WES) data of three patients in Korea (two siblings and their mother) found a novel variant in the *COCH* gene in chr14: 31349796G > A with SNHL and episodic vertigo [62]. The protein homology analysis of Cochlin (Figure 4) showed close similarity to chinchilla, followed by the mouse model. It is interesting to note that there was no homologue for *Drosophila*, and similarity with zebrafish was only 58.7%. This suggests that the protein might have evolved to its specialized role in mammals. Due to the complex structure of the organ of Corti and the specific expression of the cochlin, the mouse model would be the best model to study the functional effect of this mutation in the pathophysiology of MD.
***DPT***

The gene *DPT* (OMIM: 125597) encodes for the protein dermatopontin (Q07507), an extracellular protein with a potential role in cell matrix interactions and matrix assembly. It is highly expressed in the stomata layer of the neural tissues; however, they exhibit low expression in the human inner ear [63].

Martin-Sierra et al. [13] found a Spanish family of three sisters suffering from MD, who had a missense variant 1:168665849G > A [13]. Comparing the protein sequence of the human dermatopontin (Figure 4) revealed a close similarity to the mouse Dpt. This was followed by zebrafish, which showed a similarity of 64.2%. Hence, zebrafish could be an attractive model to study the DPT mutation in the pathophysiology of MD. Of note, non-mammal model organisms, such as Drosophila, showed very low similarity of 32.2%.
***SEMA3D***

Semaphorin-3D protein (O95025) is encoded by the gene *SEMA3D* (OMIM: 609907). Semaphorins are extracellular proteins that function as axonal guidance molecules, which are known to play a role in regulating the motility and morphology of different cell types that form nervous, immune, renal, endocrine, cardiovascular, and many more complex systems [64]. Intriguingly, SEMA3D expression in mice peaks at P1 in the organ of Corti, but drops at P7, suggesting its active role in maturation of the mouse organ of Corti [63].

Martin-Sierra et al. [13] found a novel missense variant in a Spanish family with three individuals in the same generation affected with MD. While the three segregated the variant (chr7: 84642128G > A), there were three additional individuals from different generations with incomplete phenotypes. The novel missense mutation was classified as pathogenic, with the amino acid changing from proline to serine. In addition, Gallego-Martinez et al. [65] reported an enrichment of rare missense variants in sporadic MD in genes that are involved in the axonal guidance signaling pathway. Furthermore, Requena et al. [66] compared differentially expressed genes in mice HCs and non-HCs and reported that the axonal signaling pathway plays a pivotal role in cochlear and vestibular supporting cells. To better understand the relevance of *SEMA3D* in MD, the best preclinical model would be the mouse model, which shares 92.9% of its protein sequence (Figure 4) with the human protein. Conversely, if we consider a non-mammalian model, then the zebrafish would be preferred, since it shares a similarity of 63.8% to the human semaphorin-3D.
***TECTA***

The gene *TECTA* (OMIM: 602574) encodes the α-tectorin protein (O75443), which is a major component of the tectorial membrane (TM), an extracellular matrix that lays over stereocilia of the sensory hair cells in the organ of Corti. Due to its role in gating of the mechanotransduction channels and mediation of deflection of stereocilia of the sensory hair cells during hair cells’ transduction, TM plays a pivotal role in hearing function [67]. Hence, mutations in TECTA can damage the structural integrity of the TM, potentially leading to hearing loss. As previously mentioned, *TECTA* is one of the most common genes associated with FMD. *TECTA* mutations lead to nonsyndromic SNHL in humans [68], and their potential role in MD was suggested in a mouse model (Tecta^C1509G^) that developed EH [69]. Roman-Naranjo et al. [70] reported three missense variants and two deletions in the *TECTA* gene, segregating in several families with MD. The three missense variants (11:121158016T > C, 11:121152980G > C, and 11:121165368C > T) and two deletions (11:121157956AC > A and 11:121189864GC > C) may lead to protein instability and slimming of the TM, leading to the pathogenic phenotype.

While comparing the protein sequence of *TECTA* with other model organisms, our analysis revealed a similarity of 59.6% with zebrafish tecta protein (Figure 4). Intriguingly, the similarity was in domains encompassing Zonadhesin. It is worth noting that this region has the highest number of reported variants for MD [70]. Hence, zebrafish would be a preferred model to study this gene.
***GUSB***

The *GUSB* gene (OMIM: 611499) encodes for an enzyme called glucuronidase beta (P08236). It is required to breakdown large molecules, called glycosaminoglycans (GAGs). Loss of function of *GUSB* leads to an autosomal recessive lysosomal storage disease called mucopolysaccharidosis VII (MPS VII) [71,72]. Of the many pathologies this disease manifests, hearing loss is one of them [73]. The mouse model for MPS VII revealed that SNHL was associated with alterations of the cochlear structure, accompanied by defects in the vestibular system [74]. This phenotype suggests *GUSB* as a suitable candidate for MD. In MD patients, one missense variant, 9:65980297C > T, was reported by Skarp et al. [75]. Protein sequence similarity between human and other model organisms revealed that the protein had 74.9% similarity to the mouse orthologue. This suggests that the protein was not highly conserved. Non-mammalian models that could potentially be used to study the protein are zebrafish, which share a similarity of 65.3%, followed by Drosophila with 46.5% similarity.
***SLC6A7***

The gene *SLC6A7* (OMIM: 606205) encodes for the protein Solute carrier family 6 member 7 (Q99884). It is a member of the solute carrier family, which are membrane transport proteins that transport sugars, amino acids, nucleotides, and drugs [76]. Other solute carriers expressed in the inner ear, such as SLC26A4, SLC22A4, and SLC26A5, have been well characterized [77,78,79]. Skarp et al. [75] found a rare variant for *SLC6A7* in a family with MD. The missense variant, 5:150196820G > C, c.322G > C/p.V108L, found in the family could suggest *SLC6A7* as a potential candidate gene for MD. Additionally, it is implicated in oxidative stress, making it a novel pathway to understand MD. Disruption of SLC transporters has been previously studied in guinea pig, which shares 97.8% of the protein with the human orthologue [80]. Next, the non-mammalian animal model with around 51% similarity is the zebrafish analogue, whereas Drosophila has only 44.6% similarity, making it an unsuitable candidate for a preclinical model (Figure 4).
***GJD3***

The gene *GJD3* (OMIM: 607425) encodes the Gap junction protein delta 3 (Q91YD1). Its mouse orthologue (Connexin 30.2), which shares a protein homology of 81.1%, is expressed in the TM, spiral limbus, nerve fibers, and base pole of the inner hair cells. Additionally, it is also observed in the stria vascularis, basilar membrane, inner hair cells, and outer hair cells in younger mice. This suggests its expression to be pivotal for hearing function. Escalera-Balsera et al. reported a total of 18 MD patients with a complex haplotype in the *GJD3* gene [63]. This rare haplotype (TGAGT) consisted of two synonymous, two missense, and one downstream variant. This haplotype segregated, in total, three unrelated families of ten individuals, and another eight individuals with sporadic MD [81].

Based on the protein sequence homology (Figure 4), the ideal model organism to study *GJD3* function would be zebrafish (38.5%). While it is relatively low as compared to other model organisms, such as mouse or guinea pig, it is higher than its Drosophila orthologue (35.7%). Interestingly, the conserved domain in zebrafish lies within the transmembrane domain. This domain is important for its function (passage of ions and small molecules). Hence, zebrafish would be a preferred model to study *GJD3* mutations in MD.

## 5. Genes Associated with Autosomal Recessive Familial MD

Five genes have been reported to be associated with autosomal recessive familial MD: *STRC*, *HMX2*, *TEME55*, *OTOG*, and *LSAMP.*
***STRC***

The gene *STRC* (OMIM: 606440) encodes the protein stereocilin (Q7RTU9). The gene is expressed in the inner ear and required for hearing function. It is specifically expressed in the horizontal top connectors that link outer hair cell stereocilia, attaching it to the TM in the organ of Corti [82]. It is also one of the most common targets for genetic mutations that leads to recessive SNHL (DFNB16) [83,84,85]. The *STRC* mutation carrier frequency is quite high, with some reports suggesting it to be around 2.6% in the human population [86]. While the exact frequency of *STRC* mutations is yet to be determined, a large study would enable researchers to understand the contribution of STRC mutations to hearing loss. By using exome sequencing, Frykholm et al. [87] found three members, consisting of two brothers and a first cousin, with MD. While the two brothers shared a bi-allelic mutation in *STRC* chr15: 43896948G > A (nonsense homozygous variant), the cousin had the same mutation but exhibited heterozygosis accompanied by a deletion of around 97 kb of the *STRC* gene. The inheritance pattern was autosomal recessive, as none of the parents showed any signs of the disease [87]. Mouse models for *STRC* have been well studied. In these mice, the absence of stereocilin leads to progressive hearing loss, starting at P15 [82]. After comparing the protein sequence of *STRC* with other model organisms, there was no significant difference between Drosophila (30.2%) and zebrafish (29%). Hence, both may have low homology with the human protein, and the mouse model would be preferred.
***HMX2***

The homeobox protein HMX2 (OMIM: P43687) encoded by the gene *HMX2* (600647) is a transcription factor important for specification of neuronal cell types. It is well studied for its role in hypothalamus and inner ear development [88]. The knockout mouse model exhibited a defective inner ear with complete loss of the vestibular system, accompanied by hearing loss, suggesting its pivotal role in development of the inner ear [88]. The protein is highly conserved in mice and zebrafish [89,90]. Skarp et al. [75] found a novel heterozygous missense variant (chr10: 123150118T > A (Tyr273Asn)) in a Finnish family (child and his paternal grandfather) suffering from MD. To understand the significance of this mutation in *HMX2* and study the pathophysiology of MD, zebrafish would be an ideal model. With 64.4% similarity to the protein sequence of human *HMX2* (Figure 4), zebrafish hmx2 is more homologous, as compared to the Drosophila (Hmx) protein, which shares only 42.6% similarity.
***TMEM55B***

The gene *TMEM55B* (OMIM: 609865) encodes the protein type 1 phosphatidylinositol 4,5-bisphosphate 4-phosphatase (Q86TO3). The protein is part of the TMEM family of transmembrane proteins that are important for lysosomal trafficking and cellular homeostasis [91]. It is interesting to note that TMEM132E, another member of the TMEM protein family, is associated with autosomal recessive nonsyndromic deafness (DFNB99) [92]. Skarp et al. [75] found that a Finnish family, comprising of a child and his paternal grandfather, suffering from MD with a missense mutation in the HMX2 gene also had a missense variant in TMEM55B chr14: 20459211C > T (Leu229Phe). Intriguingly, it is one of the few reported cases of childhood onset of MD, illustrating that an ideal animal model is crucial to understand the pathophysiology of MD. The protein sequence similarity for zebrafish is high (72.2%) (Figure 4), suggesting it to be an ideal candidate to study this gene in an animal model.
***OTOG***

The gene *OTOG* (OMIM: 604487) encodes for Otogelin (Q6ZR10), a secretory protein that is required for anchoring of the TM to the hair cell stereocilia in the organ of Corti and the otolithic membrane in the vestibular organs [93]. Moreover, it is also known to be involved in the stabilization and organization of TM, suggesting a potential role in the regulation of mechanotransduction. Given that the protein is expressed in the inner ear [94], and its function is critical for the normal functioning of the inner ear, its deletion in mouse models showed severe hearing and vestibular dysfunction [23,93]. Otogelin is more abundantly expressed in the vestibule that in the cochlea [24].

*OTOG* is one of the most common genes associated with FMD, with around ten missense variants discovered in Spanish families [1,95,96]. While some of the variants showed a compound heterozygous recessive inheritance, exome sequencing further revealed few rare missense variants in the *OTOG* gene, suggesting its significant role in FMD [95]. Protein sequence analysis of *OTOG* orthologues (Figure 4) revealed a similarity of 43.7% for the *OTOG* protein in zebrafish, whereas other non-mammalian animal models, such as Drosophila, shared only 30.1% similarity. The mouse model, which shared a protein homology of 82.5%, is the preferred model to study the *OTOG* gene.
***LSAMP***

The gene *LSAMP* (OMIM: 603241) encodes for the Limbic system membrane protein (Q13449), which is a neuronal surface adhesion glycoprotein in subcortical and cortical regions of the limbic system [97]. Mehrjoo et al. [98] found a novel missense variant, 1318915.2: c.673 T  > C, in two sisters, segregating the phenotype, in a consanguineous Iranian Lur family suffering from MD [98]. The lsamp protein in the mouse orthologue has three isoforms with similarity as high as 99%. Interestingly, the zebrafish orthologue of lsamp has a similarity of 61.1%, which is higher than guinea pig (56.8%) (Figure 4). Hence, zebrafish would be a preferred model organism to study lsamp in FMD.

## 6. Genes Associated with Digenic Familial MD

Five genes have been reported that co-segregate rare missense variants with *MYO7A* in several unrelated families with MD. The proteins encoding these genes have a physical interaction in the hair cell stereocilia and could work synergistically with *MYO7A*, which can cause or exacerbate the phenotype of MD.
***MYO7A***

The gene *MYO7A* (OMIM: 276903) encodes the protein Myosin VIIA (Q13402), an actin-binding motor protein highly expressed in the cochlea [99]. It is involved in the formation of stereocilia of hair cells in both the vestibular and cochlea systems [100]. It plays a pivotal role as a tip-linked motor of the stereocilia in slow adaption and in mechano-electric transduction. Furthermore, *MYO7A* has been commonly associated with FMD [101]. Several rare variants (n = 8) of *MYO7A* have been identified so far in MD patients [102]. The protein sequence homology (Figure 4) was compared with other model organisms and, surprisingly, zebrafish *MYO7A* had a similarity of 83.7%. It is interesting to note that researchers have already started utilizing the zebrafish to study *MYO7A* hearing loss [103,104]. The Drosophila orthologue, which showed 61.4% similarity to human *MYO7A*, has also been actively used to study molecular mechanisms of the gene [105]. Since both models have been well established, it would be ideal for both to be used to study variants of MYO7A with regards to studying the pathophysiology of MD.
***ADGRV1***

The gene *ADGRV1* (OMIM: 602851) encodes the adhesion G-protein-coupled receptor V1, a ubiquitous protein (Q8WXG9) that is expressed in various tissues, in addition to the inner ear [106,107,108]. The protein is part of the ankle link complex that is expressed in the base of the hair cell stereocilia [109,110]. Due to its expression pattern, it is not only important for hair cell development, but also for auditory function [111]. Recently, Roman-Naranjo et al. [102] found two rare missense heterozygous variants of *ADGRV1* (90694338: c.7582C > T and 90840606: c.16640G > A) in two families who were suffering from MD. Intriguingly, the patients also carried *MYO7A* gene variants. Mutations in *ADGRV1* are implicated in the Usher syndrome type II, a congenital disease characterized by deafness and blindness [112].

Mouse models for *ADGRV1* harboring loss of function mutations of the gene have been well characterized and have led to severe hearing deficits [111,113]. Looking at the protein sequence (Figure 4) and comparing it to other model organisms revealed that *ADGRV1* does not have a Drosophila orthologue. The ideal model organism to study would be the zebrafish orthologue, with a similarity of 52.5%. Hence, zebrafish would be the preferred model to investigate the pathophysiology of MD. Due to its relevance in Usher syndrome, Stemerdink et al. [114] generated an *adgrv1^rmc22^* zebrafish model to study the pathophysiology of Usher syndrome. Hence, establishing a zebrafish model to study the variants of *ADGRV1* with respect to MD would be a feasible option.
***CDH23***

The protein cadherin 23 (Q9H251) is encoded by the gene *CDH23* (OMIM:605516). It is an important component of the stereocilia tip link of the hair cells and essential for the arrangement of stereocilia [115,116]. Due to its expression in the stereocilia, the protein is also implicated in both nonsyndromic SNHL (DFNB12) and Usher syndrome 1D (USH1D) [117,118]. Roman-Naranjo et al. [102] found two missense variants in *CDH23* (71732116: c.3845A > G and 71793440: c.6512G > A) in two unrelated families with MD, along with rare variants in the MYO7A gene. Mouse models (both inbred and transgenic) to study the function of the mouse Cdh23 gene have been successfully established (*Cdh23*^12J^, *Cdh23*^Ahl^*, Cdh23*^CBA/CaJ^*, salsa*, and *waltzer*^v2J^) [119,120,121,122]. Looking at non-mammal models to study *CDH23*, Yang et al. [123] successfully established a zebrafish model to study congenital hearing loss by generating Cdh23 null embryos [123]. In contrast, a *CDH23* Drosophila model has not been established. It is worth noting that Drosophila Cdh23 shares only 28.9% (Figure 4) with the human cadherin 23. Nevertheless, the homology is mostly in the cadherin domain, which is important for its calcium-binding function. This suggests Drosophila as an alternative model to study the function of *CDH23* in MD.
***PCDH15***

The protocadherin-related 15 (Q96QU1) is encoded by the *PCDH15* gene (OMIM: 605514). It is expressed in the tip links of the hair cell stereocilia and is associated with Usher syndrome 1F and autosomal recessive deafness 23 [124]. The first reported case of *PCDH15/MYO7A* digenic inheritance was by Yoshimura et al. [125]. Roman-Naranjo et al. [102] reported a missense variant (53822490: c.5257C > A) in two members of a family suffering from MD. The protein sequence homology (Figure 4) with other model organisms revealed that pcdh15b (zebrafish orthologue) was 57% similar and Cad99C (Drosophila orthologue) was 28.3%. It is worth noting that zebrafish and Drosophila models for *PCHD15* have already been established to study Usher syndrome type 1 [126,127]. Hence, both models can equally be used to decipher the functional relevance of the mutation in MD.
***USH1C***

The gene *USH1C* (OMIM: 605242) encodes the Harmonin (Q9Y6N9) protein. Harmonin is an adaptor protein, along with Myosin VIIA and cadherin 23, within the sensory hair cell bundles that provides scaffolding for F-actin to connect to the upper end of the tip link of the stereocilia [128]. Variants in USH1C are phenotypically related to Usher syndrome type 1C and DFNB18A [129]. Roman-Naranjo et al. [102] reported a missense variant (17509546: c.1823C > G) in a family with MD. While this was found as a heterozygous variant in this study, the variant was previously found in a Chinese family as a homozygous variant causing Usher syndrome 1C [130]. It is postulated that the variant reported by Roman-Naranjo et al. [102], due to the presence of the *MYO7A* variant, could contribute to digenic inheritance. Due to its significance in Usher syndrome, the protein has been well studied in other model organisms. In zebrafish, Phillips et al. [131] found a role in photoreceptor synaptic development and function. The Drosophila orthologue protein sequence (Figure 4) showed 38% similarity to Harmonin, which is not high, but makes it an attractive alternative to study the *USH1C* variant for FMD.
***SHROOM2***

The Shroom family member 2 protein (Q13796) is encoded by the *SHROOM2* gene (OMIM: 300103). The function of SHROOM2 is not fully characterized but is known to be expressed in the hair cells. It has been suggested to interact with the tail of Myosin VIIA protein [132,133]. The *SHROOM2* variant reported by Roman-Naranjo et al. [102] is a missense variant (9894539: c.631G > A) found in two families who already had variants in PCDH15. This could suggest a possible interplay between *MYO7A*, *PCDH15*, and *SHROOM2* involving the mechanotransduction. To better understand the functional relevance of the *SHROOM2* variant in FMD, we examined the protein sequence homology (Figure 4) with a non-mammal model organism. We found a similar homology for zebrafish (45%) and Drosophila (31.7%). Therefore, zebrafish would be the preferred model to study SHROOM2 in FMD.

## 7. Conclusions

Understanding the pathophysiology of MD is critical for the management of this inner ear disorder. The usage of murine models has been at the forefront of neuroscience research for decades when it comes to studying disease pathology that affects the hearing and vestibular organs. In this review, we examined alternative, cost-effective animal models, such as Drosophila and zebrafish, to study the pathophysiology of MD. Each model has its own strengths and limitations when it comes to studying the inner ear. While the zebrafish model offers greater flexibility to undertake developmental studies and its forward genetic screens have been at the forefront in identifying novel deafness genes, Drosophila has its own benefits, including the rapid generation of transgenic flies and availability of abundant genetic tools for easy manipulation, making both models attractive options to study MD.

We examined genes that have been previously identified in FMD and compared the protein sequence homology across different animal models. Our analysis showed that all genes, except *ADGRV1*, *COCH*, *DPT*, and *STRC*, are relatively well conserved in Drosophila, showing significant homology with the human protein. Each gene and model organism to study a candidate mutation should be selected on a case-by-case basis. One of the factors to be taken into consideration is the homology of the protein domain that harbors the mutation. Introducing the zebrafish or Drosophila models to investigate MD mutations could pave a new pathway to better understand the molecular biology and prognosis of this disease. This would eventually help us to develop future gene-based therapies for MD.

## Figures and Tables

**Figure 1 jcm-14-01427-f001:**
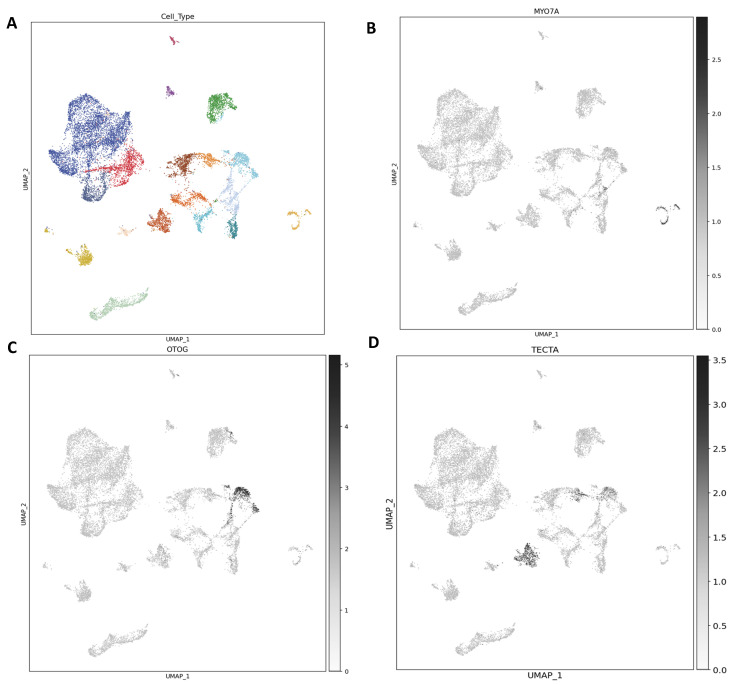
Gene expression profile of *MYO7A*, *OTOG*, and *TECTA* in the human inner ear using snRNAseq. (**A**) Different cell types that are color coded that were used for the analysis. (**B**) Expression profile of *MYO7A*. (**C**) Expression profile of *OTOG.* (**D**) Expression profile of *TECTA.* Figure and data adapted from [22].

**Figure 2 jcm-14-01427-f002:**
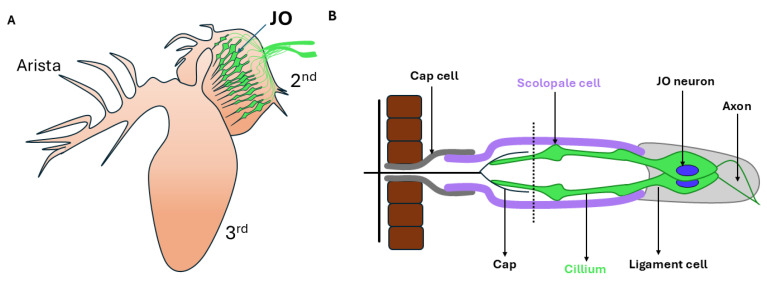
The auditory organ of the fruit fly. (**A**) Diagrammatic illustration of a fruit fly’s antenna, showing its second and third segments and the arista. The second segment harbors JO (green). (**B**) An enlarged diagrammatic representation of Johnston’s organ. Figure adapted from [35].

**Figure 4 jcm-14-01427-f004:**
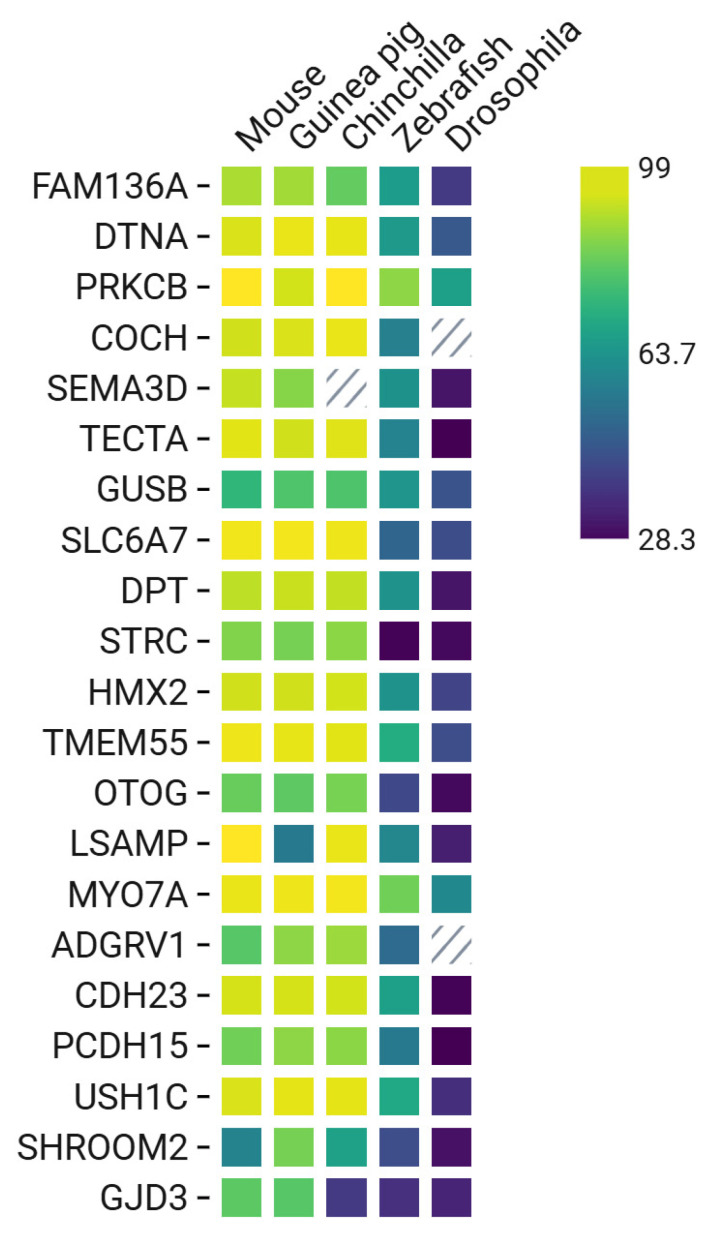
Protein homology in the 21 genes associated with Meniere’s disease in rodents, zebrafish, and *Drosophila* models. (Graph created using Bio render).

## Data Availability

No new data were created.

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
