# Peer review of "Preclinical Models to Study the Molecular Pathophysiology of Meniere’s Disease: A Pathway to Gene Therapy"

_jcm, 2025, doi:10.3390/jcm14051427_

Round 1
Reviewer 1 Report
Comments and Suggestions for Authors
Its a nicely written piece of work. I suggest to check some typo errors.
Author Response
- Its a nicely written piece of work. I suggest to check some typo errors.
Thank you for your comments we have made the necessary changes in the revised manuscript.
Reviewer 2 Report
Comments and Suggestions for Authors
This review is timely and helpful. I have a number of general comments that I list below.
The use of English is mostly adequate, except that the authors do not properly apply article adjectives (a, an, the) that should precede most nouns. This problem is everywhere.
Many citations are missing, especially after 'et al.' This also is everywhere.
There are many singular/plural problems, with mismatches between noun and verb. This also is everywhere.
There are also many cases of noun/verb mismatch that are not singular/plural; they are just the wrong noun/verb pairs. The text should be analyzed, sentence by sentence, for noun/verb agreement.
The authors seem to have no idea when to use commas. This renders the text harder to read.
Likewise, the authors seem not to understand the use of 'that' versus 'which'. These words denote different contexts. There are a lot of them. Please find them.
Comments often appear regarding whether a particular degree of sequence similarity makes a given model more or less useful. It is not just the similarity that may determine the usefulness of a model. It may also be the ability of the model to display the expected pathology. These comments need to be less simplistic.
The treatment of hydrops is not satisfying. It only appears twice in the text. What may be different about MD+hydrops, and does that make certain genes (maybe those that generate a soluble protein) better candidates for MD+hydrops versus MD-hydrops?
Please explain the coordinates in Figure 1.
Comments on the Quality of English LanguageSee above
Author Response
- The use of English is mostly adequate, except that the authors do not properly apply article adjectives (a, an, the) that should precede most nouns. This problem is everywhere.
Thank you for your comments we have made the necessary changes in the revised manuscript.
- Many citations are missing, especially after 'et al.' This also is everywhere.
Thank you for your comments we have made the necessary changes in the revised manuscript.
- There are many singular/plural problems, with mismatches between noun and verb. This also is everywhere.
Thank you for your comments we have made the necessary changes in the revised manuscript.
- There are also many cases of noun/verb mismatch that are not singular/plural; they are just the wrong noun/verb pairs. The text should be analysed, sentence by sentence, for noun/verb agreement.
Thank you for your comments we have made the necessary changes in the revised manuscript.
- The authors seem to have no idea when to use commas. This renders the text harder to read.
Thank you for your comments we have made the necessary changes in the revised manuscript.
- Likewise, the authors seem not to understand the use of 'that' versus 'which'. These words denote different contexts. There are a lot of them. Please find them.
Thank you for your comments we have made the necessary changes in the revised manuscript.
- Comments often appear regarding whether a particular degree of sequence similarity makes a given model more or less useful. It is not just the similarity that may determine the usefulness of a model. It may also be the ability of the model to display the expected pathology. These comments need to be less simplistic.
Thank you for the comment, true protein sequence similarity cannot be the only factor to consider choosing a model organism. In this review we aim to look at this as a preliminary screening to choose a model organism. This will be followed by looking at the function of the protein and the expected pathology of the protein.
- The treatment of hydrops is not satisfying. It only appears twice in the text. What may be different about MD+hydrops, and does that make certain genes (maybe those that generate a soluble protein) better candidates for MD+hydrops versus MD-hydrops?
Thank you for the comment, Endolymphatic hydrops is associated with Meniere disease and other inner ear disorders. Research has found an association between hydrops and sensorineural hearing loss in humans, and rodent models are needed if the main goal is to access endolymphatic hydrops. We have added this text to line 74.
- Please explain the coordinates in Figure 1.
Thank you for your comments we have made the necessary changes in the revised manuscript.
Reviewer 3 Report
Comments and Suggestions for Authors
In this article, the authors discuss why zebrafish and drosophila are two model organisms that are useful for studying Meniere’s Disease (MD). Mouse models and other mammal models like guinea pigs and chinchillas are useful for hearing research but they can be expensive and take a lot longer to generate data. The authors do a good job summarizing why zebrafish and drosophila are useful models for studying MD. For each of the main genes associated with MD in humans, they go into detail to what these genes do and say whether zebrafish or drosophila are models that can be used for studying these specific genes. This is an informative article that can help other researchers in the future study these genes and their role in MD. There are a few things that would be helpful to add or change to make it ready for publication. The comments and concerns are listed below. Once these comments are accurately addressed, this would be a useful article that would help other researchers in the future.
· Lines 46-47- It would be helpful to give a few more sentences about what causes MD and how the disease is characterized. I know there are many unknowns when it comes to the exact cause but briefly discussing the possible causes could be helpful to the reader. What cell types are most effected?
· Line 79- Should probably say “gEAR at different ages.” Looks like there was a typo
· For Figure 1, It would be good to switch the A and B panels. It makes more sense to have A be the different cell types that are color coded for the analysis and then have B-D be the gene expression profiles for the 3 genes of interest. This isn’t super critical, but it makes more sense to switch A and B for Figure 1.
· Line 102-104- Please reword this sentence. There seems to be some words missing
· Line 108- the text states “several hundred years during evolution.” I am guessing that is meant to say several million years or several hundred million years because several hundred years would not be correct.
· Line 113- the text states “or loss of disrupts hair cell differentiation.” Please reword
· There is no figure callout in the text for Figure 2 or for Figure 3.
· It would be helpful to have more information in the text that describes the similarities between the Johnston organ of Drosophila and the cochlea and vestibular system of mammals. How similar are the cell types between the two and more information about the functional similarity between the two would be helpful. Pointing out the similarities between the Johnston organ and cochlea of mammals would really help convince the reader that Drosophila are a useful model system for studying MD.
· It would also be helpful to have more information about the similarities between zebrafish and the cochlea of the mammals. More background about these similarities would really help convince the reader that zebrafish are a useful model for studying MD.
· Zebrafish are commonly used as a drug screening model for otoprotective compounds and that helps strengthen your case that they are useful models for studying MD. Many drugs that protect mammals from hearing loss also protect from HC loss in zebrafish. Might be helpful to mention that in the text. This shows the similarities between the hearing organ in zebrafish and the cochlea for mammals.
· The strengths and limitation tables that you have are extremely helpful and they are a great addition to the manuscript. It would be helpful to have citations for each strength and limitation. Not all the points have a citation, and it would be helpful to have a citation that supports each statement.
· Line 178- Please reword the sentence. It is confusing the way it is written.
· What are you using as the % cutoff for whether the protein homology for zebrafish and drosophila are similar enough to humans to consider it close enough to use it as a model for MD? Sometimes, it is stated that 60 something % is close enough to humans so it makes it a viable model for studying MD but sometimes the % is as low as 30 something (STRC gene) and that is good enough for using zebrafish and drosophila as MD models. Is the % homology more important or that the gene function in zebrafish and drosophila are similar to mammals? It would be helpful to state exactly what the criteria was to say whether or not zebrafish and drosophila are useful models for studying the different genes in MD.
· For the conclusion, it would be very helpful to have a few sentences that sum up the main reasons why zebrafish and drosophila are useful models for studying MD. This would really help and convince the reader why these are useful models and give them the main takeaways of what the article is trying to convey. Mention some of the biggest strengths that each model has and why they are good alternatives to mouse and other mammal models.
· In general, please go over the entire article and double check some of the grammar. There are multiple places where there is a word missing or the wrong word is used which makes some of the sentences confusing to read. Overall, it is well written, just a few things need to be addressed and a careful read over would help with the readability of the article.
Author Response
- Lines 46-47- It would be helpful to give a few more sentences about what causes MD and how the disease is characterized. I know there are many unknowns when it comes to the exact cause but briefly discussing the possible causes could be helpful to the reader. What cell types are most effected?
Thank you for your comments we have made the necessary changes ( line 44-46) in the revised manuscript.
- Line 79- Should probably say “gEAR at different ages.” Looks like there was a typo
Thank you for your comments we have made the necessary changes in the revised manuscript.
3. For Figure 1, It would be good to switch the A and B panels. It makes more sense to have A be the different cell types that are color coded for the analysis and then have B-D be the gene expression profiles for the 3 genes of interest. This isn’t super critical, but it makes more sense to switch A and B for Figure 1.
Thank you for your comments we have made the necessary changes in the revised manuscript.
4. Line 102-104- Please reword this sentence. There seems to be some words missing
Thank you for your comments we have made the necessary changes (107-108) in the revised manuscript.
5. Line 108- the text states “several hundred years during evolution.” I am guessing that is meant to say several million years or several hundred million years because several hundred years would not be correct.
Thank you for your comments we have made the necessary changes (line 120) in the revised manuscript.
6. Line 113- the text states “or loss of disrupts hair cell differentiation.” Please reword
Thank you for your comments we have made the necessary changes (line 125) in the revised manuscript.
7. There is no figure callout in the text for Figure 2 or for Figure 3.
Thank you for your comments we have made the necessary changes in the revised manuscript.
8. It would be helpful to have more information in the text that describes the similarities between the Johnston organ of Drosophila and the cochlea and vestibular system of mammals. How similar are the cell types between the two and more information about the functional similarity between the two would be helpful. Pointing out the similarities between the Johnston organ and cochlea of mammals would really help convince the reader that Drosophila are a useful model system for studying MD.
Thank you for your comments we have made the necessary changes (lines 111-118) in the revised manuscript.
9. It would also be helpful to have more information about the similarities between zebrafish and the cochlea of the mammals. More background about these similarities would really help convince the reader that zebrafish are a useful model for studying MD.
Thank you for your comments we have made the necessary changes (lines 147-153) in the revised manuscript.
10. Zebrafish are commonly used as a drug screening model for otoprotective compounds and that helps strengthen your case that they are useful models for studying MD. Many drugs that protect mammals from hearing loss also protect from HC loss in zebrafish. Might be helpful to mention that in the text. This shows the similarities between the hearing organ in zebrafish and the cochlea for mammals.
Zebrafish is used for drug screening for otoprotective compounds and for this reason this could be an excellent MD model to test novel drugs. We have added this to one of the strengths to use Zebrafish as a MD model.
11. The strengths and limitation tables that you have are extremely helpful and they are a great addition to the manuscript. It would be helpful to have citations for each strength and limitation. Not all the points have a citation, and it would be helpful to have a citation that supports each statement.
Thank you for your comments we have made the necessary changes in the revised manuscript.
12. Line 178- Please reword the sentence. It is confusing the way it is written.
Thank you for your comments we have made the necessary changes in the revised manuscript.
13. What are you using as the % cutoff for whether the protein homology for zebrafish and drosophila are similar enough to humans to consider it close enough to use it as a model for MD? Sometimes, it is stated that 60 something % is close enough to humans so it makes it a viable model for studying MD but sometimes the % is as low as 30 something (STRC gene) and that is good enough for using zebrafish and drosophila as MD models. Is the % homology more important or that the gene function in zebrafish and drosophila are similar to mammals? It would be helpful to state exactly what the criteria was to say whether or not zebrafish and drosophila are useful models for studying the different genes in MD.
Thank you for the comment. To choose the ideal animal model, we evaluate each gene on case-by-case basis. For instance, we used % homology of protein as one of the preliminary criteria. This was followed by looking at the function of the gene in the organism. So, for some genes such as STRC where the gene has already been extensively studied in mouse, we look at Drosophila or Zebrafish to better understand the gene and its function with respect to MD even though protein homology is quite low (30%). For gene such as COCH whose Drosophila homologue doesn’t exist and zebrafish homologue is (58.7 %) we choose mouse model due to the unique expression pattern of COCH and the complex structure of the mouse organ of Corti which cannot me mimicked by Drosophila or zebrafish.
14. For the conclusion, it would be very helpful to have a few sentences that sum up the main reasons why zebrafish and drosophila are useful models for studying MD. This would really help and convince the reader why these are useful models and give them the main takeaways of what the article is trying to convey. Mention some of the biggest strengths that each model has and why they are good alternatives to mouse and other mammal models.
Thank you for your comments we have made the necessary changes (Lines 458-461) in the revised manuscript.
15. In general, please go over the entire article and double check some of the grammar. There are multiple places where there is a word missing or the wrong word is used which makes some of the sentences confusing to read. Overall, it is well written, just a few things need to be addressed and a careful read over would help with the readability of the article.
Thank you for your comments we have made the necessary changes in the revised manuscript.
Round 2
Reviewer 2 Report
Comments and Suggestions for Authors
I found this article useful, but many of the same problems persist--singular/plural mismatches, misplaced commas, that/which swaps non-grammatical uses of colons and semicolons, and missing words. Also, 'comprise' is generally used two ways, as verb, or passively '...is comprised of...'. I will seek to send along my highlighted copy if the journal allows. This is just sloppy.
There do not appear to be any gene therapies for MD coming along, as the implicated genes are rare and low-effect. I would drop the reference to gene therapy in the title.
The role of hydrops is still not well-treated. The article states in line 48 that MD is considered to be a disease of endolymph regulation. Is this true? I would thus presume the authors believe all MD is hydropic to some extent? Please do a better job with this. Frankly hydropic MD may represent a specific subtype with its own genes.
Line 88. What are 'endolymphatic cells"
Figure 1 uses contradicting color codes by cell type and heat map. This is confusing.
I should have said this last time, but are zebrafish and drosophila really good MD models if it is a condition of endolymph dysregulation? What would MD look like in these models? Either be more convincing or delete these models.
The evidence for GUSB, USH1C, and SHROOM2 as candidate genes for MD is weak. I would strengthen or delete.
Line 312. The basilar membrane is acellular and does not 'express' anything.
CDH23 mutations were first discovered in inbred mouse lines where they naturally arose. Most variants listed in this section are not transgenic lines.

See above.
Author Response
- I found this article useful, but many of the same problems persist--singular/plural mismatches, misplaced commas, that/which swaps non-grammatical uses of colons and semicolons, and missing words. Also, 'comprise' is generally used two ways, as verb, or passively '...is comprised of...'. I will seek to send along my highlighted copy if the journal allows. This is just sloppy.
We sincerely apologize for the previous grammar and typo mistakes. We have revised the and made the necessary changes in the revised manuscript. All these changes have been highlighted (in red) in the R2 version.
- There do not appear to be any gene therapies for MD coming along, as the implicated genes are rare and low-effect. I would drop the reference to gene therapy in the title.
We respectfully disagree about future gene therapy for MD. This perspective article describes the pre-clinical models available to conduct the required studies to develop gene therapy for Meniere disease. There are 20 candidate genes already associated with familial MD. Particularly, rare variants in OTOG and MYO7A have been reported in multiple MD families. Moreover, these genes have been previously associated with SNHL and imbalance in KO mouse models. Currently, two mouse transgenic mouse models containing human wild type and mutated exons have been generated for OTOG and MYO7A, and gene therapy in these models is on the horizon.
- The role of hydrops is still not well-treated. The article states in line 48 that MD is considered to be a disease of endolymph regulation. Is this true? I would thus presume the authors believe all MD is hydropic to some extent? Please do a better job with this. Frankly hydropic MD may represent a specific subtype with its own genes.
We apologize for any misunderstanding. We have stated that MD is associated with an accumulation of endolymph in the cochlear duct. This finding is a consequence of the inner ear damage, and it is associated with the hearing loss threshold. So far, no gene has been associated with endolymphatic hydrops, which is a histopathological finding also observed in other disorders different from MD.
- Line 88. What are 'endolymphatic cells"
Thank you for the question. Transitional or endolymphatic cells are cells termed according to scRNAseq expression data obtained from human inner ear organoids. We have modified the lines 89-90 regarding the expression of OTOG in the organ of Corti.
- Figure 1 uses contradicting color codes by cell type and heat map. This is confusing.
The data shown in figure 1 was extracted through a publicly available data exploration platform hosted by the Expression Analysis Resource: https://umgear.org/p?l=hIEOandInnerEar. We have modified the gene expression heatmaps for MYO7A, OTOG and TECTA to a grey to black scale.
- I should have said this last time, but are zebrafish and drosophila really good MD models if it is a condition of endolymph dysregulation? What would MD look like in these models? Either be more convincing or delete these models.
The reviewer is assuming that MD is a consequence of endolymph dysregulation, but this is not the case in all cases. Genetically modified Drosophila and Zebrafish models can be good animal models to demonstrate hearing loss and vestibular dysfunction, which can be measured using auditory-evoked responses and vestibular behavior (S Baeza-loya et al 2023, TT Whitfield et al 2002, PR Senthilan et al 2012, B Cellini 2024).
- The evidence for GUSB, USH1C, and SHROOM2 as candidate genes for MD is weak. I would strengthen or delete.
We fully agree that the evidence for some candidate gene is limited to human genetic data generated from a single family, but deleting genes pending confirmation in other families will not help to validate these genes. Further human and animal studies are needed.
- Line 312. The basilar membrane is acellular and does not 'express' anything.
We have changed “expressed” by “Additionally, the protein shows immunolabelling in the stria vascularis, basilar membrane,..” on line 314.
- CDH23 mutations were first discovered in inbred mouse lines where they naturally arose. Most variants listed in this section are not transgenic lines.
You are right. We have modified the line 427 to : “…Mouse models (both inbred and transgenic) to study function of mouse Cdh23 gene have been successfully established.
Thank you for your comments we have made the necessary changes in the revised manuscript.